# Natural Variation in Plant Pluripotency and Regeneration

**DOI:** 10.3390/plants9101261

**Published:** 2020-09-24

**Authors:** Robin Lardon, Danny Geelen

**Affiliations:** Department of Plants and Crops, Horticell, Ghent University, 9000 Gent, Belgium; robin.lardon@ugent.be

**Keywords:** callus formation, de novo organogenesis, epigenetics, mapping, natural variation, plant regeneration, pluripotency, quantitative trait loci, tissue culture

## Abstract

Plant regeneration is essential for survival upon wounding and is, hence, considered to be a strong natural selective trait. The capacity of plant tissues to regenerate in vitro, however, varies substantially between and within species and depends on the applied incubation conditions. Insight into the genetic factors underlying this variation may help to improve numerous biotechnological applications that exploit in vitro regeneration. Here, we review the state of the art on the molecular framework of de novo shoot organogenesis from root explants in *Arabidopsis*, which is a complex process controlled by multiple quantitative trait loci of various effect sizes. Two types of factors are distinguished that contribute to natural regenerative variation: master regulators that are conserved in all experimental systems (e.g., *WUSCHEL* and related homeobox genes) and conditional regulators whose relative role depends on the explant and the incubation settings. We further elaborate on epigenetic variation and protocol variables that likely contribute to differential explant responsivity within species and conclude that in vitro shoot organogenesis occurs at the intersection between (epi) genetics, endogenous hormone levels, and environmental influences.

## 1. Introduction: Definition, Origin, and Applications of Regeneration

As sessile organisms, plants face numerous environmental stresses, and, accordingly, they have retained extended developmental plasticity compared to animals [1]. This ability to redirect the fate of differentiated somatic cells potentiates tissue repair and organ reconstruction after injury during postembryonic growth and de novo formation of various plant structures from in vitro explant cultures upon exposure to phytohormones or abiotic stresses, which is collectively termed regeneration and has contributed to several biotechnological applications [2,3]. For instance, various ornamentals are produced by micropropagation, in which excised tissues are cultured on growth media supplemented with plant hormones to regenerate whole plantlets that can be transferred to soil. This ability to recreate the entire body from a few cells, or even a single cell, is commonly known as totipotency, and it was first postulated by Haberlandt in 1902, although it was not until 1957, when Skoog and Miller discovered that the ratio of auxin and cytokinin in the culture medium determines the identity of newly formed organs, that the first regeneration protocols were established [4]. The advantages of the approach include a large propagation factor, while the fixation of strong genotypes because of asexual reproduction and extensive control over the culture conditions ensures high-yielding phenotypes [5]. On the other hand, genetic homogeneity should be carefully monitored to avoid somaclonal variations that might arise during subculture, unless these are desirable for selection purposes [6]. Other applications are virus sanitization and the development of transgenic crops, which requires regeneration from tissues or protoplasts transformed with a particular genetic construct. When applied to pollen, this leads to the creation of haploid plants that can be chemically converted to homozygous individuals for use in crop breeding [7,8]. Most differentiated tissues, however, are limited to pluripotency, i.e., the reconstruction of specific cell types. This is exploited in techniques such as cutting and grafting, allowing the reuse of resilient root stocks with high-yielding scions. The success of these approaches is highly variable between species, cultivars, and even explants of the same individual and we are only beginning to understand the sources of such variation. This review summarizes various regeneration systems, along with recent advances in our understanding of the molecular framework underlying de novo shoot organogenesis in *Arabidopsis thaliana*. Based on that knowledge, we attempt to bridge the gap between reports on organogenic variability between and within different species and natural variation in several regulatory layers of regeneration.

## 2. Different Regeneration Systems

While simple organisms such as liverworts and mosses are able to reconstruct their full body without additional hormonal signals after injury [9], regeneration in seed plants involves a complex interplay of multiple inductive cues to repair cuts or recreate organs [10]. Cereals and woody species often show recalcitrance to tissue cultivation and micropropagation due to restricted availability or accessibility of pluripotent stem cells [7]. Three key systems have been used to study and exploit plant regeneration in *Arabidopsis thaliana* (Figure 1) [1,2]. The first is based on an extensive capacity for wound repair and organ reconstruction that allows for root tip regeneration after laser ablation of the quiescent center (QC) or excision of the root apical meristem (RAM), the repair of stem tissues after incision and reconnection of the vasculature of stock and scion during grafting, all relying on populations of competent cells in the adult plant body and polar auxin transport [2,10]. Besides tissue repair, the two most common types of regeneration in higher plants are somatic embryogenesis and de novo organogenesis, which can both occur in a direct or indirect way depending on the requirement for an intermediate callus stage (Figure 1) [1,11]. Somatic embryogenesis reinitiates embryonic developmental sequences in (de)differentiated cells to generate bipolar structures with a clear distinction between root and shoot meristems [1,2,8,12]. This can be induced by abiotic stresses such as salt, heavy metals, heat, and drought [13,14], but it is usually achieved by treatment with the synthetic auxin 2,4-dichlorophenoxyacetic acid (2,4-D) via the formation of embryogenic callus [8,12]. The subsequent transfer of explants to hormone-free medium initiates somatic embryogenesis, creating cellular structures reminiscent of globular, heart-shaped, and torpedo-shaped embryogenic stages through the coordinated action of transcription factors (TFs) such as LEAFY COTYLEDON (LEC) 1 and 2, AGAMOUS-LIKE 15 (AGL15), FUSCA 3 (FUS3), BABYBOOM (BBM), and EMBRYOMAKER (EMK) [5,8,12]. De novo organogenesis, on the other hand, refers to the formation of new meristems from pluripotent stem cells to reconstruct organs [2]. Typical examples include adventitious rooting from detached leaves, petioles, or hypocotyls (also termed rhizogenesis) and shoot regeneration from in vitro cultured root or hypocotyl segments (sometimes referred to as caulogenesis) [10]. The outcome of both processes is governed by the plant hormones auxin and cytokinin as a high auxin ratio favors root growth, while high cytokinin content promotes shoot fate [4]. De novo root regeneration has been exploited for vegetative propagation through cuttings and can be achieved by cultivation on auxin-rich callus-inducing medium (CIM), followed by transfer to root induction medium (RIM) with less or no auxin [2,10]. More recent systems dispense with external hormone supply, resulting in direct root regeneration governed by wounding, endogenous hormone levels, and a transcriptional cascade involving *WUSCHEL-RELATED HOMEOBOX* (*WOX*) *11* and *12*, *WOX5&7*, and *LATERAL ORGAN BOUNDARIES DOMAIN* (*LBD*) *16* and *29* [15,16,17]. Similarly, the two-step protocol for de novo shoot organogenesis, in which explants are preincubated on auxin-rich CIM before transfer to shoot induction medium (SIM) with high cytokinin levels, constitutes a vital step in many transformation protocols [4,10,18]. Aside from hormone signaling, shoot regeneration involves wound responses for callus formation from founder cells and acquisition of organogenic competence, followed by transdifferentiation of root-like protuberances into shoot primordia, patterning of the shoot apical meristem (SAM), and subsequent organ outgrowth [4,7,19,20,21,22]. Cellular and molecular events underlying these stages are discussed in the next section.

## 3. Cellular and Molecular Framework of de novo Shoot Organogenesis

### 3.1. Auxin and Cytokinin Signalling

Auxin and cytokinin play vital, but often antagonistic roles in de novo shoot formation and mutations in the homeostatic or signaling pathway of either hormone are known to impair regeneration [4,23]. Auxin is synthesized from tryptophan (Trp) in a two-step reaction orchestrated by TRYPTOPHAN AMINOTRANSFERASE (TAA) and YUCCA (YUC) enzymes, and its perception involves an SKP1-cullin-F-box (SCF)-type E3 ubiquitin ligase, in which an F-box protein from the TRANSPORT INHIBITOR RESISTANT 1 (TIR1)/AUXIN SIGNALING F-BOX (AFB) family provides substrate specificity [24]. In the presence of auxin, this SCF^TIR1/AFB1-5^ coreceptor complex binds Aux/IAA (INDOLE-3-ACETIC ACID INDUCIBLE) repressor proteins and targets them for proteasomal degradation, which releases TOPLESS (TPL)-mediated repression of AUXIN RESPONSE FACTORs (ARFs) that induce auxin-responsive gene expression. Cytokinins, on the other hand, are synthesized from adenosine triphosphate (ATP) or diphosphate (ADP) in two successive reactions, respectively catalyzed by ISOPENTENYLTRANSFERASE (IPT) and LONELEY GUY (LOG) enzymes [24]. Active hormone levels are also regulated by conjugation and irreversible degradation, mainly by cytokinin oxidases or dehydrogenases (CKX) [4]. Cytokinin is perceived by a multicomponent His-to-Asp phosphorelay, starting at one of three hybrid ARABIDOPSIS HISTIDINE KINASEs (AHKs) that autophosphorylate upon cytokinin binding and transfer the P_i_ to ARABIDOPSIS HISTIDINE PHOSPHOTRANSFER PROTEINs (AHPs). The latter act as shuttle proteins, conveying the signal into the nucleus, where they activate two types of ARABIDOPSIS RESPONSE REGULATORS (ARRs): type-B ARRs that act as transcriptional activators of the cytokinin response, and type-A ARRs that form a negative feedback loop on the signal [4]. Auxin and cytokinin show extensive crosstalk during shoot regeneration, exemplified by reciprocal control over each other’s biosynthesis [25,26]. Ethylene (ET), brassinosteroids (BR), gibberellin (GA), and abscisic acid (ABA) interfere in the process as well [19,23].

### 3.2. Wound Responses

Intact seedlings subjected to the CIM–SIM procedure regenerate lateral roots instead of shoots, indicating the importance of wound stress for de novo SAM formation [27]. Moreover, new organs are often formed naturally at cut sites, and wounding enhances several types of hormone-induced regeneration, including somatic embryogenesis [7,10]. Early wound responses involve rapid Ca^2+^ influxes, plasma membrane depolarization, a burst of reactive oxygen species (ROS), disruption of cellular communication, and jasmonic acid (JA) accumulation, although it is not well understood how these are translated into waves of extensive transcriptional changes observed after wounding [7,28]. Central regulators of wound-induced reprogramming are the APETHALA2/ETHYLENE RESPONSE FACTOR (AP2/ERF) TF WOUND-INDUCED DEDIFFERENTIATION 1 (WIND1) and its homologs WIND2-4, which are expressed locally within 1 h after wounding and promote cell dedifferentiation and proliferation [29,30]. Recently, it was reported that WIND1 acts by promoting cytokinin signaling through B-type ARRs and directly activating ENHANCER OF SHOOT REGENERATION 1 (ESR1) to upregulate CUP-SHAPED COTYLEDON 1 (CUC1), an important determinant of SAM formation during embryogenesis [31]. Moreover, induction of IPT3 and LOG1, LOG4, and LOG5 upon cutting elevates cytokinin levels, in turn reactivating the cell cycle via CYCLIN D3;1 (CYCD3;1) [7]. Other AP2/ERFs, such as ERF115 and PLETHORA (PLT) 3, -5, and -7, are also expressed during wound-induced callus formation, and the AP2/ERF and CK-mediated pathways undergo extensive crosstalk [28]. Many chromatin remodeling factors, including POLYCOMB REPRESSIVE COMPLEX 2 (PRC2) components and DNA methyltransferases MET1 and CHROMOMETHYLASE 2 (CMT2), are differentially regulated by cutting as well. Accordingly, it was shown that chromatin modifications undergo dynamic changes upon wounding and accumulation or loss of histone 3 lysine 4 trimethylation (H3K4me3), and lysine 9/14/27 acetylation (H3K9/14/27ac), dependent on HISTONE ACETYLTRANSFERASE OF THE GNAT FAMILY 1 and 3 (HAG1&3), is respectively correlated to transcriptional activation or repression [32]. Genes with high levels of these permissive histone marks before or shortly after wounding (such as *WIND1*, *ERF113/RAP2.6L,* and *LBD16*) tend to be rapidly induced by cutting, whereas genes with H3K36me3 and H3K27me3 are less responsive to wounding. Notably, it has been suggested that wound-induced calli differ from calli formed on CIM because the former do not express root markers and their induction is not affected in *solitary root* (*slr*) mutants [29,33,34].

### 3.3. Founder Cell Specification

When explants are placed on CIM, the high auxin concentration triggers cell division to initiate the formation of a proliferating cell mass or callus [4,35]. While it was first thought that any somatic cell can dedifferentiate and reenter the cell cycle, this ability is, in fact, restricted to populations of partially differentiated stem cells distributed throughout the adult plant body [11]. In the case of shoot regeneration from root explants, pluripotent pericycle cells, opposite of the xylem poles, are required for callus formation [35,36]. Similar to lateral root initiation, founder cells are specified by local auxin maxima instated by the AUXIN RESISTANT 1 (AUX1) and AUX1-LIKE (LAX) 1-3 influx carriers [4]. Use of the synthetic auxins 2,4-D and 1-naphthalene acetic acid (NAA) improves the efficiency of CIM because these compounds cannot be exported by PIN-FORMED (PIN) proteins and are poorly metabolized. Auxin-induced degradation of IAA28 then promotes the expression of GATA23 to confer founder cell identity. Mitotic competence of pericycle cells also relies on ABERRANT LATERAL ROOT FORMATION 4 (ALF4), the knockout of which impairs both lateral root formation and shoot regeneration [37]. Intriguingly, repression of *ALF4* by very-long-chain fatty acids confines the capacity of pericycle cells to form callus [38].

### 3.4. Callus Formation

In-vitro-induced callus resembles root primordia on a morphological and transcriptional level, even when derived from aerial organs [36,39,40]. Indeed, transcriptome comparison revealed similar expression profiles in lateral root tips and organogenic calli, which show organized expression of root meristem markers such as *WOX5*, *SHORT ROOT* (*SHR*), *SCARECROW* (*SCR*), *PLT1&2*, *PIN1*, *QUIESCENT CENTER 25* (*QC25*), *ROOT CLAVATA HOMOLOG 1* (*RCH1*), and *GLABRA2* (*GL2*). Accordingly, callus formation on CIM proceeds through a similar developmental program as lateral root initiation [7,20,34]. Auxin accumulation in the founder cells mediates degradation of IAA14/SLR, which releases ARF7 and ARF19 that upregulate LBD16-18 and LBD29, involving ATXR2-mediated H3K36me3 deposition [4,41,42]. JUMONJI C DOMAIN-CONTAINING PROTEIN 30 (JMJ30) also associates with the ARF-ATXR2 complex to promote *LBD* expression by removing repressive H3K9me3 marks [43], while the BRASSINOSTEROID INSENSITIVE 2 (BIN2) kinase integrates temperature sensitivity into this cascade by enhancing the transcriptional activity of *ARF7&19* and *LBD* genes [44]. In turn, LBD18 reinforces the auxin signal by promoting ARF7&19, and, together with LBD33, it triggers cell proliferation via transcriptional activation of *E2 PROMOTER BINDING FACTOR a* (*E2Fa*), which associates with DIMERIZATION PARTNERs (DPs) to stimulate DNA replication genes [7,11]. Auxin also downregulates KIP-RELATED PROTEIN (KRP) 2, -3, and -7, which are cyclin-dependent kinase (CDK) inhibitors, through a reduction in PROPORZ1 (PRZ1)-deposited H3K9ac and H3K14ac [34]. *KRPs* are further silenced via H4R3 dimethylation and alternative splicing of *RELATED TO KPC1* (*RKP*) by PROTEIN ARGININE METHYLTRANSFERASE 5 (PRMT5) [45]. A series of anticlinal and periclinal divisions then lead to the formation of a dome-shaped protuberance [4,35], which is further assisted by LBD-mediated cell wall modifications. For instance, LBD18 directly induces EXPANSIN 4 (EXP4), LBD29 targets PECTIN METHYLESTERASE 2 (PME2), and association of LBD16 with bZIP59 activates a FAD-binding Berberine enzyme to oxidize monolignols in the cell wall [7,46,47]. Upon proliferation of the callus, expression of the pericycle marker J0121 diffuses, concomitant with the acquisition of root identity [20]. Cytokinin interferes in callus formation via ARR7&15 and ARR1&21, which are, respectively, A-type and B-type response regulators with a negative and positive effect on the outcome, and it has been proposed that CK-control of the cell cycle is moderated by ESR1&2 [34,48]. These AP2/ERF TFs can directly induce CYCD1;1 and OBF BINDING PROTEIN 1 (OBP1), in turn activating CYCD3;3 and shortening the G1 phase to enable cell cycle reentry. The interplay between auxin and cytokinin during callus initiation is at least partly directed by microRNAs, as miR160 can repress the process by targeting ARF10 to relieve the direct suppression of ARR15 [49].

### 3.5. Pluripotency Acquisition

On CIM, calli also acquire competence to respond to shoot inductive cues as activation of key shoot meristem regulators like *WUSCHEL* (*WUS*) on SIM requires several days of CIM preincubation [35]. Recently, it was found that transient expression of root stem cell maintenance genes, including *WOX5*, *PLT1&2*, *SCR,* and *SHR*, in a subset of callus cells confers pluripotency, but how these are activated is not completely understood [11,19,50]. One pathway involves the induction of PLT3, PLT5, and PLT7 by auxin, followed by direct upregulation of PLT1 and PLT2, as well as the shoot determinants CUC1 and CUC2 [51]. The latter have been put forward as markers of pluripotency acquisition, along with AHK4, IAA20, and ARABIDOPSIS CRINKLY 4 (ACR4) [4]. Notably, the WOX11-LBD16 pathway involved in de novo root regeneration from leaf explants also contributes to the lateral root primordium (LRP) character of callus on CIM [52]. Furthermore, the acquisition of shoot competence depends on cell cycle reentry and progressive epigenetic changes [21,35]. For instance, HAG1 catalyzes the acetylation of histone H3 at *WOX5&14*, *PLT1&2,* and *SCR* loci to potentiate their transcription on CIM [50]. During callus formation from leaf blades, histone deacetylation by HDA3&9, as well as genome-wide changes in H3K27me3 levels installed by the PRC2 components CURLY LEAF (CLF), SWINGER (SWN), and EMBRYONIC FLOWER 2 (EMF2), are required to silence leaf-specific gene expression, while removal of H3K27me3 at loci involved in auxin signaling and root development enables leaf-to-callus transition [53,54]. Many other regeneration determinants, such as *WOX5&11*, *WUS,* and *SHOOT MERISTEMLESS* (*STM*), undergo dynamic changes in PRC2-mediated H3K27me3 during organogenesis, but the machinery underlying histone demethylation is still poorly characterized [10,55]. However, LYSINE-SPECIFIC DEMETHYLASE 1-LIKE 3 (LDL3) was found to remove H3K4me2 during callus formation to prime the activation of SAM-patterning genes (e.g., *ARR12*, *WUS,* and *CLAVATA3* (*CLV3*)) on SIM [56]. Besides histone modification, DNA methylation provides another mechanism of epigenetic reprogramming, and it was proposed that reactivation of the cell cycle by auxin, followed by continued cell division in the presence of high cytokinin levels on SIM, causes a dilution in the methylation status of key regeneration genes [57]. Accordingly, mutation of *CMT3* and *DOMAINS REARRANGED METHYLTRANSFERASE* (*DMR*) *1* and *2* allows to bypass CIM preincubation for *WUS* expression and shoot regeneration.

### 3.6. Transdifferentiation

It has been proposed that de novo establishment of the shoot apical meristem on SIM reflects transdifferentiation from root to shoot identity instead of a true de- and redifferentiation process [36,39,58]. After all, dedifferentiation has often been vaguely inferred from renewed cell division and morphological changes, which do not necessarily reflect reversal to an embryonic state. Its role in animal regeneration systems is currently under reconsideration as well because blastema formed during limb regeneration in salamanders were found to comprise a heterogenous pool of progenitors that regenerate tissues within their original lineage [11,58]. Adult mammalian cells also appear to retain the potential for transdifferentiation, as ectopic expression of a few TFs can induce lineage reprogramming without reversal to a stem cell identity. Similarly, LRP in plants can be directly converted into shoot primordia during a narrow developmental window that supports repetitive reversal of the organogenetic program by wavered application of 2-isopentenyladenine (2-iP) and NAA [59]. Intriguingly, the switch is paralleled by reduced root marker expression and upregulation of shoot regulators or vice versa, resulting in altered cell division patterns after a mitotic pause. Transcriptome comparison revealed that LRPs undergoing direct conversion resemble regenerating calli and both processes depend on the rapid rearrangement of complementary auxin and cytokinin domains, as well as *WOX5*-expressing stem cells, for transition [59].

### 3.7. Shoot Promeristem Formation

Upon transfer to SIM, overlapping signaling domains of auxin and cytokinin diverge into mutually exclusive regions, thereby partitioning cellular identity through complementary expression of CUC2 and WUS [7,22,25,60]. These developmental regulators initially show broad activity throughout the callus, after which CUC2 is restricted to areas with high auxin levels and rapidly dividing cells, whereas WUS is confined to cells marked by AHK4 and strong cytokinin responses [4,11]. As discussed before, *CUC1&2* are upregulated by PLT3, PLT5, and PLT7 during CIM preincubation, but also by ESR1&2 in response to cytokinin and wound-induced WIND1 [31,51,61]. Their combined expression activates STM in a surrounding ring of cells and modulates polar localization of PIN1 to direct the auxin flow towards the apical tip [4,11,19]. Subsequently, STM is expressed throughout the shoot promeristem and restricts *CUC* genes to the peripheral zone, where they induce LIGHT SENSISTIVE HYPOCOTYLS (LSH) 3 and 4, which suppress differentiation in the organ boundary [60,62]. On the other hand, WUS has emerged as a master regulator of stem cell maintenance in the SAM as overexpression causes ectopic shoot formation, while *wus* mutants fail to regenerate [19,21,26]. This homeobox TF is directly activated by cytokinin via the B-type response regulators ARR1, ARR2, ARR10, and ARR12, that physically interact with HD-ZIP III TFs PHABULOSA (PHB), PHAVOLUTA (PHV), and REVOLUTA (REV), to spatially confine WUS expression to shoot progenitor cells [26,63,64,65]. Oddly, mutation of *ARGONAUTE 10* (*AGO10*) was found to enhance pro-SAM formation by releasing *miR165&166* to degrade *HD-ZIP III* mRNAs [66], while this would be expected to hamper ARR-mediated *WUS* transcription, in agreement with other reports showing that *phb phv rev* and inducible miR165/166 lines fail to regenerate [26]. Moreover, HD-ZIP III TFs stimulate other shoot determinants such as *STM* and *RAP2.6L* [7,67,68], and B-type ARRs also promote WUS indirectly by repressing YUC1&4-modulated auxin biosynthesis [64]. In turn, WUS reinforces cytokinin responses by suppressing A-type ARRs, while further downregulating auxin-induced root markers by rheostatic gating of the entire auxin pathway through direct transcriptional repression and TPL-assisted association with histone deacetylases [69,70,71]. Conversely, direct suppression of *IPT5* by ARF3 and inhibition of STM-mediated *IPT7* activation maintains auxin signaling at the peripheral and apical zone of the primordium [4,25]. *WUS* is only induced after 2–3 days on SIM because it harbors the repressive H3K27me3 mark maintained by PRC2 components CLF and SWN, removal of which depends on cytokinin-controlled cell cycle progression and possibly involves active demethylation by EARLY FLOWERING 6 (ELF6) and RELATIVE OF EARLY FLOWERING 6 (REF6) [26,53]. The *WUS* locus also undergoes dynamic changes in the levels of other repressive histone modifications (e.g., H3K9me3) and activating marks (e.g., H3K9ac and H3K4me3), which is modulated by LDL3 and JMJ14 (H3K4 demethylases), KRYPTONITE (KYP; an H3K9 methyltransferase) and HAG1 [56,72]. Moreover, DRM1&2, CMT3, and MET1 normally silence *WUS* via DNA methylation, which is gradually lost on SIM [57]. It was shown that cytokinin first upregulates *MET1* through CYCD3-mediated activation of E2Fa to prevent early *WUS* transcription, after which *MET1* expression migrates to the outer layers of the callus and the resulting dilution of DNA methylation enables *WUS* activation by ARRs [73]. WUS is additionally regulated by intrinsic factors such as miR156, transcript levels of which decrease in aging plants, leading to the accumulation of SPL9-group proteins that bind B-type ARRs to attenuate cytokinin signals [19,74]. Ultimately, *WUS* is confined to the center of the dome-shaped promeristem when *CUC1&2* activity shifts to the peripheral zone and *STM* is expressed throughout the primordium [11,60].

### 3.8. SAM Patterning and Shoot Outgrowth

A functional SAM consists of an organizing center (OC) between a lower rib meristem and a central zone with slowly dividing stem cells at the apex, surrounded by a peripheral zone undergoing rapid proliferation [75,76]. While the OC maintains the stem cell niche, daughter cells of the central zone are displaced sideways into the peripheral zone or downwards into the rib meristem, where they respectively differentiate into lateral organs (i.e., leaves or flowers) or contribute to stem formation [4]. The balance between the meristematic regions is governed by WUS, which is produced in the OC and migrates to the surrounding layers to induce *CLV3*, in turn blocking *WUS* expression in the central zone and the rib meristem through CLV1&2 and CORYNE (CRN) [4,75]. This feedback loop is reinstated during shoot regeneration as *CLV3* is upregulated after *WUS* expression is established in the center of the shoot promeristem [60]. In this last stage, PIN1 orientation is shifted to the incipient leaf primordia by PINOID (PID) to redistribute auxin maxima and enable phyllotactic patterning of the meristem [4]. PIN1 and PID are further regulated by interaction with ESR1&2, and the influx carriers AUX1 and LAX1-3 were proposed to contain auxin in the epidermal cell layer, which is marked by *ARABIDOPSIS THALIANA MERISTEM LAYER 1* (*ATML1*) [60,77,78]. REVOLUTA (REV) and FILAMENTOUS FLOWER (FIL) determine adaxial/abaxial polarity in the newly formed organ primordia. Notably, the chromatin modifier ARABIDOPSIS TRITHORAX 4 (ATX4) is essential for shoot identity determination as it promotes the expression of *ARABIDOPSIS THALIANA HOMEOBOX GENE 1* (*ATH1*), *KNOTTED1-LIKE HOMEOBOX GENE 4* (*KNAT4*), *SAWTOOTH* (*SAW*) *1* and *2*, *TCP10,* and *YABBY5* (*YAB5*) by H3K4me3 deposition [19,79].

## 4. Mapping Natural Variation in the Organogenic Potential at the Genetic Level

### 4.1. Variation in Arabidopsis thaliana

Several studies have reported substantial variation in the regenerative potential of natural *Arabidopsis thaliana* accessions, and various strategies have been applied to map such differences at the genetic level (Table 1; [80]) [81,82]. However, early attempts suffered from limited mapping resolution and often failed to identify causative genes because of poor annotations. For instance, variation among recombinant inbred lines (RILs) between Col and L*er* was linked to one common region (85–89 cM) on chromosome 1 underlying callus formation and shoot regeneration from leaf explants, whereas several areas on chromosomes 4 (e.g., 24–32 cM) and 5 (e.g., 36–40 cM) specifically contributed to shoot regeneration from roots or leaves [83]. Combined with transgressive segregation (i.e., the observation that some descendant RILs showed more extreme phenotypes than the parental lines), the authors concluded that regeneration is determined by multiple loci acting at various stages and that genetic requirements also depend on the explant type. While no specific genes were pinpointed, the marker on chromosome 1 was found to be near *SHOOT REDIFFERENTIATION DEFECTIVE 1* (*SRD1*; a gene underlying temperature-sensitive redifferentiation from roots to shoots), although it also overlaps with *RECEPTOR-LIKE PROTEIN KINASE 1* (*RPK1*; 91 cM on chromosome 1), which was later discovered by Motte et al. [81,83,84]. Curiously, composite interval mapping with the Col x L*er* RILs also revealed quantitative trait loci (QTLs) on chromosomes 1, 4, and 5, but at different positions (respectively around 12.5, 60.9, and 106.7 cM) [85]. Binning lines that carried the superior Col allele or the L*er* variant of the major QTL on chromosome 5, while fixing the beneficial L*er* alleles at the minor loci and contrasting gene expression patterns in both genotype pools uncovered 845 differentially expressed genes, including *ARR18*, *AGL6,* and *AT4G36590* (2 MADS-box TFs), *AT5G50820* (a *NO APICAL MERISTEM* family member related to *CUC1&2*), *HOMEOBOX-LEUCINE ZIPPER PROTEIN 17* (*HB17*), and 3 subtilases (*AT1G01900*, *AT5G59120,* and *AT4G26330*). On the other hand, analysis of L*er* x Cvi RILs respectively identified 8, 5, and 4 loci for shoot, root, and callus formation from leaf or root explants [82]. Only a minority of QTLs were shared between different explant types and regeneration protocols (e.g., 115–130 cM on chromosome 1, 40–60 cM on chromosome 2, and 10–30 cM and 70–95 cM on chromosome 5), again indicating the occurrence of common determinants acting besides a multitude of context-dependent factors. One shoot-specific QTL (0–20 cM on chromosome 1) was proposed to represent allelic variants of *SRD3*, *ROOT REDIFFERENTIATION DEFECTIVE 4* (*RRD4*), *CYTOKININ HYPERSENSITIVE 1* (*CKH1*), *INCREASED ORGAN REGENERATION 1* (*IRE1*), or *AUXIN RESISTANT 1* (*AXR1*) [82].

More recently, linkage mapping with RILs between Nok-3 and Ga-0 revealed five major QTLs responsible for the difference in de novo shoot organogenesis between these lines [81]. One of these QTLs was refined by local association analyses based on 88 accessions, which pinpointed a single nucleotide polymorphism (SNP) in the gene encoding RPK1, altering the conformation of its putative ligand-binding domain. *RPK1* was found to be expressed in dividing pericycle cells during CIM preincubation and loss-of-function impaired de novo shoot formation [81]. This leucin-rich repeat (LRR) receptor-like kinase is involved in ABA signaling and is accordingly required for abiotic stress tolerance, but it also interferes in embryonic patterning and formation of cotyledon primordia, supported by its importance for PIN1 localization and *WOX5* expression [86,87,88]. Notably, the RPK1 homolog RPK2 regulates *WUS* expression to control SAM maintenance, and, although this function is not conserved in RPK1 [89], the two proteins show redundancy during embryogenesis and their sequences mainly differ in the kinase domain [81,86], indicating that they might simply respond to different external cues. Nonetheless, SAM-less monocotyledonous *rpk1* mutants are capable of expressing shoot markers and occasionally regenerate ectopic shoot meristems [90]. Motte et al. further showed that early and late regeneration characteristics are poorly correlated, indicating that the shoot organogenetic process can be inhibited at several intermediate stages and chlorophyll maturation does not ensure successful shoot determination [11,81]. Another recent study describing the effects of thioredoxin-mediated ROS homeostasis on shoot regeneration from root explants reported that 6 SNPs in *DCC1* (encoding a thiol-disulfide oxidoreductase family protein with a conserved DxxCxxC motif) were strongly linked to regenerative variation among 48 ecotypes [91]. DCC1 localizes to the mitochondria of the inner callus cells, where it reduces CARBONIC ANHYDRASE 2 (CA2; a subunit of the respiratory chain NADH dehydrogenase complex I) by direct interaction. Mutation of either *DCC1* or *CA2* caused increased ROS levels by reducing the activity of complex I, in turn hampering shoot regeneration by downregulating essential genes for callus and shoot promeristem formation (e.g., *WOX5&11*, *KNAT1-2&4*, *WUS*, *CLV3*, *STM,* and *CUCs*) and lowering the expression of auxin biosynthesis and signaling genes (e.g., *TAA1*, *YUC2*, *YUC4*, *YUC5*, *YUC9*, *GRETCHEN HAGEN 3.6* (*GH3.6*), and *SMALL AUXIN UPREGULATED 51* (*SAUR51*)), potentially via redox modification of PHV and TCP15 [91]. Moreover, mutation of the two missense SNPs in *DCC1* abolished the interaction with CA2 and, accordingly, the superior haplotype was associated with lower ROS levels. It is important to note, however, that while *DCC1* may be correlated with variation in shoot regeneration, it might not reflect a rate-limiting factor in natural populations as it was not discovered by mapping (i.e., no other loci were considered in this analysis).

Notably, a poor overlap is observed between QTLs obtained by linkage mapping using Nok-3 x Ga-0, L*er* x Col, or L*er* x Cvi RILs, highlighting the complex polygenic nature of regeneration (Table 1) [81,82,83,85]. Moreover, it is likely that variation within individual RIL sets partially depends on specific factors, and the use of different explants, protocols, and environmental conditions for regeneration can alter the phenotypic distribution and thus affect the underlying QTLs. Just recently, we performed a full-scale genome-wide association study (GWAS) to dissect variation in shoot regeneration from roots and several related in vitro traits among 190 natural *Arabidopsis* accessions under different conditions [80]. In agreement with the genetic complexity reported before, this analysis uncovered a myriad of quantitative trait genes (QTGs), including ARFs and ARRs, MYB and AP2/ERF2 family TFs, miRNAs, receptor-like kinases, F-box proteins, chromatin remodelers, and various biosynthetic and cell wall modifying enzymes, the majority of which were protocol and trait-specific (~ 95%), leading us to hypothesize that shoot regeneration is governed by a multitude of conditional fine-tuning factors and a few universal master regulators. A shift in rate-limiting QTGs under different settings could also explain the heterogeneous results obtained in other surveys. Plotting the number of associated phenotypes against a score for prior links with organogenesis based on literature allowed us to distinguish four categories of candidate genes, showcasing *WUS* as a major determinant of regeneration. We showed that its expression correlates with the regeneration potential and found numerous other a priori candidates with more specific functions (e.g., *IAA9*, *ARF4*, *ARR2*, *LSH4*, *CLE2*, *MYB118*, *FUS3*, *miR393A,* and *miR394A*). Moreover, our GWAS also exposed several novel QTGs that might act as master regulators, including *AT3G09925* (a pollen Ole e1 allergen and extensin family gene), *SUPERMAN* (*SUP*), *EMBRYO SAC DEVELOPMENT ARREST* (*EDA40*), and *DNA-BINDING WITH ONE FINGER 4.4* (*DOF4.4*). Finally, the genetic architecture of in planta shoot growth was shown to be complex as well, comprising around 112 QTLs grouped in hotspots on chromosomes 1, 2 and 5, of which ~10% were considered major QTLs (e.g., *CRYPTOCHROME 2* (*CRY2*)), and the rest were loci with intermediate or minor effects that occasionally interacted with drought stress [92]. Many of the linked regions differed across RIL sets, showed epistatic effects, and contained multiple subpeaks.

**Table 1 plants-09-01261-t001:** Overview of QTL mapping studies on regeneration in *Arabidopsis thaliana*. The QTL and QTG columns respectively specify the number (and chromosomal position) of quantitative trait loci and highlight quantitative trait genes (plausible candidates are bracketed).

*Species*	*Population*	*Method*	*Phenotype(s)*	*QTL(s)*	*QTG(s) *	*Reference*
*Arabidopsis thaliana*	Col x L*er* RILs	MAPMAKER QTL analysis	Callus and shoot formation from root or leaf explants	9 (chr 1, 4, and 5)	(*SRD1*)	Schiantarelli et al. 2001 [83]
	Col x L*er* RILs	Composite interval mapping	Shoot regeneration from root explants	3 (chr 1, 4, and 5)	*ARR18, AGL6, HB17, AT4G36590, AT5G50820, AT1G01900, AT5G59120, AT4G26330*	Lall et al. 2004 [85]
	L*er* x Cvi RILs	QTL interval mapping	Shoot or root regeneration from root or leaf explants	17 (chr 1, 2, and 5)	(*SRD3, RRD4, IRE1, CKH1, AXR1*)	Velázquez et al. 2004 [82]
	Nok-3 x Ga-0 RILs	Linkage and association mapping	Shoot regeneration from root explants	5 (chr 1–3)	*RPK1*	Motte et al. 2014 [81]
	48 ecotypes	Linkage disequilibrium analysis	Shoot regeneration from root explants	1 (chr 5)	*DCC1*	Zhang et al. 2018 [91]
	170 natural SALK strains	GWAS	Shoot regeneration from root explants	86 (chr 1–5)	*WUS, SUP, AT3G09925, DOF4.4, EDA40, QUL2, URH1, RLP9, QKY, ARF20, MSL3, DREB1A, WAVH2, MIR393A, …*	Lardon et al. 2020 [80]

### 4.2. Variation in Other Species

#### 4.2.1. Monocot Crops

The creation of transgenic rice plants for breeding and functional genome research relies on *Agrobacterium*-mediated transformation of in vitro cultures, and, although efficient protocols have been established for callus induction and regeneration from mature seeds, many commercial cultivars (e.g., Koshihikari) are recalcitrant to this system [93,94]. Like in *Arabidopsis*, early endeavors to map variation in several callus induction traits by following the segregation of amplified/restriction fragment length polymorphism (AFLP/RFLP) markers after crossing strongly and poorly regenerating (typically *japonica* and *indica*) varieties uncovered a myriad of QTLs across the 12 chromosomes, some of which could be used for marker-assisted selection, but failed to identify specific genes (Table 2) [93,95,96,97]. However, Nishimura et al. applied map-based cloning to refine one of four QTLs underlying the differential regeneration abilities of Koshihikari and Kasalath strains, pinpointing a locus on the short arm of chromosome 1 (45.4 cM), designated *PROMOTER OF SHOOT REGENERATION 1* (*PSR1*) [98]. Cloning of all four genes in the region suggested that *PSR1* is a putative ferredoxin-nitrite reductase with lower expression and activity in poorly regenerating Koshihikari plants. Another study monitored simple sequence repeat (SSR) markers in chromosome segment substitution lines (CSSLs) between Nipponbare and Zhenshan 97B to identify 29 QTLs for 4 callus traits under 2 protocols, which were grouped in hotspots on chromosomes 1, 3, and 10 [99]. More recently, high-quality genetic maps, including SNPs among two different RIL sets, uncovered largely non-overlapping batches of 8 and 25 QTLs (highlighting two major loci on chromosomes 3 and 7), most of which had not been discovered before [94,100]. The dissimilarity in QTLs was attributed to variable culture conditions, parental lines, and mapping approaches. Moreover, phenotypic contributions usually range between 6% and 26% (while older QTLs likely explain more variation because they contain multiple subassociations), implying comparable genetic complexity as in *Arabidopsis* [101]. Accordingly, a GWAS on 510 sequenced rice accessions revealed 88 loci correlated to the rate, speed, and time of callus induction, 21 of which were located in previously reported QTLs [102]. Among the candidate genes, three were orthologs of callus formation genes in *Arabidopsis*: *CROWN ROOTLESS 1* (*CRL1*; orthologous to *LBD17/29*), *OsBBM1* (an ortholog of *BBM*), and *OsSET1* (a SET-domain-containing gene, orthologous to *SWN*). While the latter was downregulated during in vitro culture, expression levels of the former two increased, consistent with the repressive and promotive roles of their respective counterparts in *Arabidopsis*. Fourteen additional candidates were put forward based on *p*-values, annotations, and expression patterns, including a putative thioredoxin, two AP2-domain-containing proteins, and OsIAA10, RNAi knockdown of which impeded callus induction and auxin responsivity [102]. Lastly, an allele of *BROWNING OF CALLUS 1* (*BOC1*) from wild rice was found to reduce callus browning in *indica* cultivars by decreasing cell death and senescence in response to oxidative stress [103].

QTL mapping of regeneration traits has been attempted in several other cereal crops such as maize [104,105,106], wheat [107,108,109], and barley [110,111,112] (Table 2). Early studies in maize using the highly and lowly regenerable A188 and B73 inbred lines identified a major QTL on chromosome 3 [104] that was recently exploited to create a germplasm with enhanced tissue culture responses, nearly isogenic to the elite B73 [106]. Fine mapping of this locus revealed a 3053-kb region containing multiple candidate genes such as *ALDOLASE 1* (*ALD1*), *ZmWOX2A,* and *ZmWOX5B*. Furthermore, a multilocus GWAS for five embryonic callus traits using 43427 SNPs in 144 inbred maize lines identified 63 common quantitative trait nucleotides (QTNs), 15 of which were retained in multiple environments [105]. A total of 40 candidate genes were found, including TFs and kinases involved in auxin transport, cell fate specification, seed germination, embryo development, and transgenic callus regeneration. Particular attention was given to *WOX2*, showing elevated expression in a strong regenerator, which is consistent with the role of *WUS* in *Arabidopsis* organogenic variation [80] and the observation that immature maize embryos transformed with *WUS2-* or *BBM*-expressing constructs regenerate more seedlings [113]. In wheat, DArT (diversity arrays technology)-assisted linkage mapping of callus induction and regeneration from mature embryos in RILs between a synthetic hexaploid (SHW-L1) and a commercial cultivar (Chuanmai 32) uncovered 6 QTLs on chromosomes 1A&D, 3B, 4&5A, and 6D, explaining up to 12% of phenotypic variation each and confirming the importance of group 1, 3, and 5 chromosomes established by previous QTL analyses on immature embryos and microspore cultures [108,109,114]. Notably, many loci were novel and only detected in one growing season, again highlighting the effect of the environment on rate-limiting regeneration determinants. Three major loci for transformation amenability were discovered in barley, and it was later shown that introgression of the corresponding alleles from Golden Promise plants into recalcitrant cultivars improves transformation efficiency [115]. Higher density mapping of these QTLs with SNP markers suggested that they might be linked to the barley homologs of *WUS2* and *BBM* [116]. Finally, a barley map based on expressed sequence tags enabled the detection of 8 regions associated with green or albino plant regeneration [112]. Four of these overlap with previously identified QTLs (on chromosomes 2H, 3H, 6H and 7H), and underlying genes were linked to hormone biosynthesis and signaling, cell cycle regulation, chloroplast maturation, and shoot meristem development. Specific examples include a ferredoxin–nitrite reductase, cyclins and CDKs, ET biosynthetic enzymes, AP2 TFs, and orthologs of *STM*, *PICKLE* (*PKL*), *LEC*, *AGL24,* and *CUCs*.

**Table 2 plants-09-01261-t002:** Nonexhaustive overview of QTL mapping studies on regeneration in monocot crops since 2000. The QTL and QTG columns respectively specify the number (and chromosomal position) of quantitative trait loci and highlight quantitative trait genes (plausible candidates are bracketed).

*Species*	*Population*	*Method*	*Phenotype(s)*	*QTL(s)*	*QTG(s) *	*Reference*
*Oryza sativa*	Norin 1 x Tadukan F_2_	QTL interval mapping	Shoot regeneration from mature seed-derived callus	2 (chr 2 and 4)	-	Takeuchi et al. 2000 [96]
	Milyang 23 x Gihobyeo RILs	RFLP analysis	Shoot regeneration from mature seed-derived callus	6 (chr 1–3 and 11)	-	Kwon et al. 2001 [97]
	Koshihikari x Kasalath BC_1-3_F_2_	Map-based cloning	Shoot regeneration from mature seed-derived callus	4 (chr 1–3 and 6)	*PSR1/NIR*	Nishimura et al. 2005 [98]
	Koshihikari x Kasalath BC_1_F_3_ and NILs	MAPMAKER QTL analysis	Callus induction and regeneration from mature seeds	8 (chr 1, 4, and 9)	-	Taguchi-Shiobara et al. 2006 [93]
	Nipponbare x Zhenshan 97B CSSLs	Stepwise regression of CSSLs	Callus induction and regeneration from mature seeds	29 (chr 1, 3, and 10)	-	Zhao et al. 2009 [99]
	Nipponbare x 93-11 RILs	Composite interval mapping	Callus induction and regeneration from mature seeds	25 (chr 3 and 7)	-	Li et al. 2013 [94]
	Pei’ai 64s x Yangdao 6 RILs	QTL mapping	Callus induction from mature seeds	8 (chr 5–7, 9, and 10)	-	Tian et al., 2013 [100]
	510 natural accessions	GWAS	Callus induction from mature seeds	88 (chr 1–12)	*OsBBM1, OsSET1, OsIAA10, CRL1, …*	Zhang et al. 2019 [102]
	Teqing x YIL25 F_2_	Map-based cloning	Callus browning	1 (chr 3)	*BOC1*	Zhang et al. 2020 [103]
*Zea Mays*	A188 x B73 NILs	Segregation distortion analysis	Callus induction and regeneration from mature embryos	1 (chr 3)	*ALD1, ZmWOX2A, ZmWOX5B*	Salvo et al. 2018 [106]
	144 inbred lines	GWAS	Callus induction and regeneration from immature embryos	63 (chr 1–10)	*ZmWOX2, ZmOCL5A, ZmBR2, ZmKIP1, ZmDEK35, ZmSBP18, …*	Ma et al. 2018 [105]
*Triticum aestivum*	Wangshuibai x Nanda 2419 RILs	Simple and composite interval mapping	Callus induction and regeneration from mature embryos	13 (chr 2A, 2D, 5A, 5B, and 5D)	-	Jia et al. 2007 [108]
	Svilena x Jensen F_3_	DArT-based QTL mapping	Green plantlet regeneration from microspore cultures	2 (chr 1B and 7B)	-	Nielsen et al. 2015 [114]
	Chuanmai 32 x SHW-L1 RILs	DArT-based QTL mapping	Callus induction and regeneration from mature embryos	6 (chr 1A, 1D, 3B, 4A, 5A, 6D)	-	Ma et al. 2016 [109]
*Hordeum Vulgare*	Azumamugi x Kanto Nakate Gold RILs	Composite interval mapping	Callus induction and shoot regeneration from immature embryo cultures	8 (chr 1–3H, 5H, and 7H)	*UZU, SHD1, VRS1*	Mano et al. 2002 [111]
	Steptoe x Morex DHs	EST-based linkage mapping	Green and albino plant regeneration	8 (chr 1–7H)	*HvSTM, HvPKL, HvLEC, HvBBM, HvESR1, HvCUC, HvAGL24, …*	Tyagi et al. 2010 [112]
	Haruna Nijo x Golden Promise F_2_	Segregation distortion analysis	Transformation amenability from immature embryos	10 (chr 1–6H)	(*HvNIR*)	Hisano et al. 2016 [115]
	Full Pint x Golden Promise DHs	Segregation distortion analysis	Transformation amenability from immature embryos	3 (chr 2–3H)	*HvBBM, HvWUS2*	Hisano et al. 2017 [116]

#### 4.2.2. Dicot Crops

The shoot regeneration capacity of tomato is believed to be inherited in a dominant way, but it is not clear how many genes are involved (Table 3). Analyzing a population obtained by crossing wild *Lycopersicon peruvianum* with cultivated *L. esculentum* plants using morphological and RFLP markers pinpointed a QTL near the middle of chromosome 3, designated *Rg-1*, and proposed to act together with one or two other loci in the control of shoot regeneration from root explants [117]. Subsequently, the *Rg-2* allele from *L. chilense* was mapped to the same region and linked to an acid invertase gene [118]. More recently, interval mapping with two populations derived from *L. esculentum* and *L. pennellii* identified 6 QTLs on chromosomes 1, 3, 4, 7 and 8, underlying shoot regeneration from leaf disks and mapping of prior candidates *Rg-2* and *LESK1* (a serine/threonine kinase upregulated during shoot induction), suggested that the QTL on chromosome 3 might be another allele of *Rg-2*, which was dubbed *Rg-3* [119]. *LESK1* was not located in a QTL but other regions contained serine/threonine kinases, histidine kinases, AP2/ERF TFs, cyclins, and MADS-box genes. Notably, only the two main QTLs on chromosomes 1 and 7 were found for all three investigated traits, and only the *Rg-3* locus overlapped with previous findings based on root explants, confirming the occurrence of conditional regeneration determinants in tomato. On the other hand, *Rg-1* appears to be a master regulator as it increases both shoot and root formation, rescuing the dual effects caused by the knockout of the DELLA protein PROCERA (PRO) and the specific phenotypes of auxin-resistant *diageotropica* (*dgt*) and *lateral suppresser* (*ls*) mutants, while overexpression of *MOUSE EARS* (*ME*) only increased shoot induction on SIM [120]. In cucumber, QTL mapping uncovered four loci on chromosomes 1, 3, and 6, explaining 9.7–16.6% of the variation in cotyledon regeneration in RILs (Table 3) [121]. GWAS detected 18 SNPs in a region on chromosome 1, underlying multiple media compositions, which were linked to three candidate genes including a homolog of *Arabidopsis J3* (a chaperone that regulates H^+^ ATPases in the plasma membrane by inactivating PKS5, integrates flowering signals and causes ABA hypersensitivity and enlarged meristems when defective). This gene was upregulated in highly regenerable genotypes, and overexpression enhanced the performance of the recalcitrant RIL parent [121]. Composite interval mapping of the regeneration rate in microspore cultures of radish identified five QTLs and five corresponding candidate genes, homologous to *PRC2* subunits and flavin-binding monooxygenases with a role in auxin biosynthesis and de novo root formation (Table 3) [122]. Finally, two QTLs were discovered for protoplast regeneration in broccoli [123], and three unlinked genes were proposed to control shoot regeneration from leaf explants of wild potato (Table 3) [124].

#### 4.2.3. Ornamentals and Commodity Crops

Rapid multiplication by in vitro shoot regeneration is especially useful for ornamentals, such as roses, GWAS of which detected 88 SNPs for direct shoot organogenesis from petioles, including 20 that were shared between two traits and 12 that were linked to known candidates in *Arabidopsis* (Table 3) [125]. These were factors related to morphogenesis, epigenetic regulation, and hormone signaling, such as a GT2-like trihelix TF, an LRR receptor-like kinase, a RAP2.7-like ERF, a DNA methylation 3-like protein, a BIG auxin transporter, a GAI-like DELLA, a KNOTTED1-like 3 homeobox, and YAB2. When mapped to a strawberry genome, the QTLs fell into small clusters containing homologs of *WUS*, *CUC1*, *SOMATIC EMBRYGENESIS RECEPTOR-LIKE KINASE 1* (*SERK1*), and *RPK1*. Sunflower, on the other hand, is an important crop for oil production, and because mass clonal propagation using cultured explants would facilitate genetic engineering, a RIL population has been developed to screen for molecular markers linked to shoot regeneration from cotyledon explants and somatic embryogenesis from epidermal layers [126,127]. This revealed transgressive segregation and identified 13 and 11 QTLs, explaining 52–67% and 48–89% of the respective traits, although no candidate genes were proposed (Table 3). Poplar is another valuable commodity crop (e.g., for timber, plywood, pulp, and paper) that has also been proposed as a model system for woody species. Association mapping of callus formation from parenchyma cells in 280 poplar genotypes uncovered 8 QTGs, and a transcriptional network indicated that these were coexpressed with a multitude of cell cycle components, as well as homologs of *LBD16*, *LEC1&2*, *WUS*, *TSD1,* and *CLF* (Table 3) [128]. Another recent study reported 83 loci linked to 275 QTGs underlying in vivo shoot traits in poplar [129].

**Table 3 plants-09-01261-t003:** Nonexhaustive overview of QTL mapping studies on regeneration in dicot crops since 2000. The QTL and QTG columns respectively specify the number (and chromosomal position) of quantitative trait loci and highlight quantitative trait genes (plausible candidates are bracketed).

*Species*	*Population*	*Method*	*Phenotype(s)*	*QTL(s)*	*QTG(s) *	*Reference*
*Lycopersicon esculentum*	L. *chilense* x KOT BC_1_F_2_	Bulked segregant analysis	Shoot regeneration from root explants	1 (chr 3)	*RG-2/INVCHI*	Satoh et al. 2000 [118]
	S. *pennellii* x Anl27 F_2_ and BC_1_	QTL interval mapping	Shoot regeneration from leaf disks	6 (chr 1, 3, 4, 7, and 8)	*RG-3, (LESK1, ESR1&2)*	Trujillo-Moya et al. 2011 [119]
*Cucumis sativus*	9110 Gt x 9930 RILs and 115 core accessions	QTL interval mapping and GWAS	Shoot regeneration from cotyledon explants	4 (chr 1, 3, and 6)	*ATJ3*	Wang et al. 2018 [121]
*Raphanus sativus*	GX71 x GX50 F_1_	Composite interval mapping	Somatic embryogenesis from microspore cultures	5 (chr 3, 8, and 9)	*PRC2, FLAVIN-BINDING MONO-OXYGENASE, SET DOMAIN PROTEIN*	Kim et al. 2020 [122]
*Brassica oleracea*	CGC 3-1 x Daehnfelt 360-7 F_2_	Simple interval mapping	Shoot regeneration from protoplast-derived calli	2 (-)	-	Holme et al. 2004 [123]
*Rosa sp.*	96 cultivars	GWAS	Direct shoot regeneration from leaf petioles	88 (chr 1 and 3–6)	*GT2-like, RAP2.7-like, MET3-like, KNOX1-like 3, ANP1-like, GAI-like, YAB2, BIG, WUS, CUC1, SERK1, RPK1, …*	Nguyen et al. 2017 [125]
*Helianthus annuus*	PAC-2 x RHA-266 RILS	Composite interval mapping	Shoot regeneration from cotyledon explants	13 (chr 2, 6–9, 15, and 17)	-	Flores Berrios et al. 2000a [127]
	PAC-2 x RHA-266 RILS	Composite interval mapping	Somatic embryogenesis from epidermal layers	11 (chr 1, 3, 4, 6, 11, 13, and 15–17)	-	Flores Berrios et al. 2000b [126]
*Populus trichocarpa*	280 genotypes	GWAS and co-expression analysis	Callus formation from parenchyma cells	8 (chr 3, 4, 6, 8, 9, 12, 15, and 18)	*SOK1, MAPK3, SCR-like, RALF-like, WUS, LBD16, LEC1&2, CLF, TSD1, …*	Tuskan et al. 2018 [128]

## 5. Other Sources of Regenerative Variation

### 5.1. Epigenetic and Transcriptional Variation

Because many key histone modifiers and DNA methylation enzymes, as well as their target marks and genes, are differentially regulated during incubation on CIM and SIM, epigenetics constitute an important regulatory layer of de novo shoot organogenesis [7,40,53,130]. Moreover, several such factors were suggested to underpin regeneration QTLs, and many important TFs and meristem determinants contain SNPs in the promoter region rather that the gene body [80], indicating that sequence variation in the epigenetic machinery and corresponding transcriptional changes might be preferred for fine-tuning gene activity over alterations in protein conformation. In turn, this implies that natural epigenomic and transcriptomic variation could be at the base of regenerative differences. With respect to this, it is important to note that genome-wide methylation patterns in *Arabidopsis* can be stably inherited over generations and are subject to selection in the context of rapid evolutionary responses [131,132,133]. Accordingly, the methylomes of 1028 natural accessions are highly variable, and many loci are polyepiallelic, meaning that they can be either unmethylated, show transposable element (TE)-like methylation (teM; mCG, mCHG, or mCHH linked to transcriptional repression), or gene body methylation (gbM; mCG normally coupled to constitutive gene expression, but also promoting the development of silenced teM alleles) [134,135]. Whereas TEs and sequence repeats are consistently and densely methylated by small interfering RNA (siRNA)-directed maintenance [133,136], the extent of gbM, in particular, differs across 725 strains and is inversely correlated with heterochromatin methylation levels through a feedback loop involving CMT3, H3K9me2, and histone turnover [137]. On the other hand, teM interacts with allele-specific gene imprinting, and it has been suggested that siRNAs contribute to epigenetic differences between strains as well [138,139]. GWAS on differential methylation highlighted many prior candidates, such as *CMT2*, *AGO1&9,* and *NUCLEAR RNA POLYMERASE D1B* (*NRPD1B*; involved in RNA-directed DNA methylation), while the observation that eQTLs and epi-eQTLs (respectively SNPs and differentially methylated regions (or DMRs) underlying gene expression) were often found at transcription start sites indicates that the general silencing effect of methylation is mediated by altered TF binding [134]. This also follows from the large overlap between DMRs and the (epi)cistrome, although eQTLs and epi-eQTLs appear to target largely distinct TF sets (e.g., *CUC2* binding sites were specifically enriched at eQTLs). Notably, epiRILs (RILs with similar DNA sequences but diverging methylomes) show comparable phenotypic inheritance as RILs or natural accessions [131], confirming that epigenetic variation contributes to environmental adaptation [132,134]. These epiRILs have also been used to map DMRs explaining variability in plant growth, morphology, and plasticity, revealing around 20 epigenetic QTLs, of which 8 had pleiotropic effects (i.e., these were required for multiple traits under different conditions) [140]. Hence, epiRILs and methylome-wide association studies (MWAS) provide interesting future prospects for elucidating additional regulatory layers of shoot regeneration [141]. In addition to DNA methylation, it was reported that H3K27me3 levels show allele-specific inheritance, but only a few targets differ between Col and L*er* strains, which has been coupled to the spreading of repressive heterochromatin marks (e.g., H3K9me2) from nearby TE insertions [142]. Although the underlying mechanisms are not straightforward, variable methylation and histone modification patterns amplify transcriptomic variation, underlined by differential expression of 22,085 genes among 998 natural accessions, which further drives phenotypic variation and adaption [134,135,137,142]. Transcriptome-wide association studies (TWAS) could be applied to identify eQTLs for de novo shoot organogenesis.

### 5.2. Source of the Explant and Hormone Responsivity

Explant age, physiological state, and source tissue (root, hypocotyl, leaf, petiole, inflorescence, microspore, etc.) determine the efficiency of regeneration, concordant with different medium requirements for callus, root, and shoot formation from root, hypocotyl, and leaf explants of diverse ecotypes [11,82,84,143]. Given the plasticity of plant morphology and inaccuracy of human work, it is impossible to ensure that such variables are perfectly uniform in the explant population. After all, many growth and developmental traits such as rosette shape, leaf expansion, and flowering time are subject to natural variation, and at any given time, different plants are at distinct developmental stages [92,144,145]. Generally, juvenile explants regenerate better than adult organs, and it has been suggested that the differential plasticity of various tissues is partly determined by changes in hormone responsivity, which may be mediated by *IRE* genes [10,146]. For instance, adventitious rooting in pea is negatively impacted by the switch from vegetative to reproductive growth, which was attributed to altered auxin homeostasis and delayed JA accumulation [147]. Accordingly, exogenous auxin application and increased cytokinin supply can respectively rescue reduced root and shoot regeneration from older *Arabidopsis* leaves [16,74]. Intriguingly, it was shown that root hormone concentrations differ between natural accessions, which was especially true for cytokinin ribosides and glucosides [148]. While moderate alterations in auxin levels and responsivity were correlated with complex root architecture, increased ethylene sensitivity was observed in poorly regenerating ecotypes [149]. Variable hormone homeostasis could also explain part of the large fluctuations in physiological and transcriptional auxin responses among strains, as these were not the result of SNPs in signaling components [150]. Furthermore, tissue-specific methylome and transcriptome patterns caused by spatiotemporal variation in endogenous hormone concentrations and cell division rates contribute to cellular heterogeneity in the callus, in turn allowing some green foci to form complete shoots when others halt their development [20]. Several additional factors have been put forward as determinants of regenerative variability within individual plants, including expression gradients of *PLT2* underlying lower repair potential at the proximal root end [151], an intrinsic timer involving *miR156* and *SPL9* that is responsible for progressively declining shoot regeneration in older plants [74] and the abundance of pre-existing adult stem cells in the explant [11].

To determine whether the capacity for shoot formation from root explants is linked to other characteristics of the parent plant, we performed a correlation meta-analysis between 16 regeneration traits in 170 *Arabidopsis* strains [80] and 383 public phenotypes from the AraPheno database that were scored in at least 50 overlapping accessions (Figure 2) [152,153]. In total, 78 of these were associated to at least one regeneration trait (at a false discovery rate (FDR) < 0.3), and 28 yielded a Spearman’s *rho* with an absolute value greater than 0.3 and an FDR below 10%. Contrary to previous reports, we detected significant positive correlations between shoot numbers, shoot primordia, undefined structures, and green callus area under two regeneration protocols, differing in the age of source plants, applied light spectrum, and external cytokinin concentration [81]. Root-like structures were comparable across protocols but form a separate category from the other traits. Intriguingly, the majority of linked public phenotypes relate to the endemic climate of the accessions under study [154], suggesting a link between regeneration and adaptation. For instance, the average monthly precipitation at the origin of strains in November and December (clim-prec11&12) is positively correlated to regeneration and especially to shoot primordia under protocol a, whereas potential and actual evapotranspiration in January (clim-pet1 and clim-pet1) are mainly connected to shoot characteristics under protocol b. Winter temperatures (e.g., clim-tmax from November through February and clim-tmin for December) show slightly positive links with shoot and callus induction, but they are inversely related to root traits, which further benefit from the inclusion of June or July in the growing season (clim-gs6&7). Root-like structures are also associated with lateral root densities and length of the basal rhizosphere under salt stress (e.g., LRDpMR0&125 and Basal0&75) [155]. Finally, shoot regeneration is negatively correlated to external characteristics of source plants, including their diameter after 5 weeks and the occurrence of rolled leaves after 8 weeks [145], as well as the concentration of metabolites with a mass-over-charge ratio of either 130 or 216 and a respective retention time of 666 or 665 s [156], suggesting a possibility for predictive models.

### 5.3. Environmental Influences

It is likely that part of the regenerative variation among explants from the same individual or subspecies can be attributed to fluctuations in environmental conditions, such as light, temperature, osmotic stress, and medium composition. For instance, it was found that variable light exposure of cotyledon explants during the first 5 days after excision affects shoot regeneration in *Arabidopsis*, and this response was different in 4 accessions [157]. More specifically, high fluorescent light intensities and early dark–light shifts impeded callus formation and regeneration, which involved the UV-A photoreceptor CRY1 and photo-oxidative damage caused by ROS accumulation. On the other hand, 2–24 h of darkness after injury and ROS quenching by xanthophylls improved shoot regeneration. The red/far-red receptors PHYTOCHROME A&B (PHYA&B) and the downstream TF ELONGATED HYPOCOTYL 5 (HY5) suppressed light inhibition by upregulating a chalcone synthase for the production of photoprotective anthocyanin pigments, which may be further enhanced by cytokinin. HY5 also mitigated the inhibitory effect of polar auxin transport, while ethylene respectively had positive and negative impacts in darkness and light [157]. Notably, calli were found to regenerate better under continuous low light exposure than using either constant high light, a 16/8 hour light–dark cycle, or complete darkness [158]. Low light intensity was also reported to promote shoot proliferation in cotton [159], and appropriate ratios of blue and red light can enhance regeneration traits in different poplar genotypes [160]. Curiously, the effects of photoperiod, light quantity, and spectrum are not always consistent between species as light is required for shoot morphogenesis in petunia [161], whereas it is not essential in tomato (although altered light–dark cycles do affect direct shoot regeneration) [162]. The use of light-emitting diodes (LEDs) could provide a cost-effective solution to address specific requirements for light intensity and wavelength in different systems [6]. Similar paradoxical observations were made for temperature effects, as both a 3-day cold pretreatment and incubation at high temperatures can increase callus formation in *Arabidopsis*, indicating the existence of an optimum for every system [44,158]. Extended cold treatments of 3–5 weeks at 3 °C could also overcome growth arrest in subcultures of crosses between peach and almond trees [163]. Optimal conditions for osmotic-stress-induced somatic embryogenesis from shoot tips and leaf buds in *Arabidopsis* differed between ecotypes [13], and the addition of sorbitol to the culture medium promoted shoot organogenesis from seed-derived callus in rice [164]. The latter relied on polar auxin transport, and highly regenerable calli were characterized by high sugar content and elevated transcript levels of invertase, sucrose transporter, *OsPIN1*, and *LATE EMBRYOGENESIS ABUNDANT 1* (*OsLEA1*) genes, implicating interaction between auxin, ABA, and carbohydrate metabolism for osmotic stress regulation in callus. Notably, constitutive osmotic stress in plastids of *Arabidopsis* MscS-like mechanosensitive ion channel (*msl2 msl3*) mutants leads to callus development at the shoot apex, involving perception of elevated cytokinin levels by AHK2, upregulation of *WUS,* and downregulation of *ARR7&15* [165]. A second independent pathway underlying this phenomenon required ROS accumulation in the SAM and increased ABA biosynthesis, which mediated retrograde signaling via downstream factors ABA INSENSITIVE 4 (ABI4) and GENOMES UNCOUPLED 1 (GUN1) to fine-tune proliferation. In *Lathyrus*, on the other hand, polyethylene glycol (PEG)-induced osmotic stress reduced the multiplication rate and vigor of regenerated shoots [166], and NaCl had opposite effects on calli of *Arabidopsis* and *Thellungiella* [167]. Finally, parameters related to the culture medium, such as nutrients (e.g., carbon source, minerals, and vitamins), plant growth regulators, solidifying agents, and pH may be regarded as constants within a particular protocol, but they can add to variation between (sub)species and tissues [143,164,168]. In *Arabidopsis*, leaf or root explants respectively prefer sucrose or glucose [143], cytokinin omission benefits protoplast development [169], activated charcoal promotes regeneration from anther-derived cultures [170], and supplementing SIM with ABA improves de novo shoot formation from roots [171].

## 6. Implications and Future Perspectives

Taken together, regeneration is a highly complex trait controlled by multiple intertwined regulatory layers that are all subject to variation, making it difficult to predict in vitro culture responses. Although significant progress has been made in elucidating the molecular framework and many potential QTLs have been identified, only a subset was mapped to specific genes. These can be divided into conditional factors and master regulators, posing the question of which established meristem determinants have specific roles in *Arabidopsis* and which ones are conserved, such as *WUSCHEL-RELATED HOMEOBOX* genes, AP2/ERF transcription factors like BBM, and receptor-like protein kinases. Knowledge of which genes are rate-limiting in a particular system can be used to overcome recalcitrance or improve the regeneration efficiency through genetic engineering. In order to avoid deleterious side-effects, the best strategies for ectopic gene activation involve the use of tissue-specific or inducible promoters (e.g., incorporating dexamethasone or estradiol-responsive elements), transient expression (e.g., based on cotransfection and segregation or inefficient T-strand processing in *Agrobacterium*-mediated methods), or excision of genes at a later stage (e.g., using *FLP* or *CRE* recombinases) [172]. Adding morphogenic genes to expression cassettes for transformation (rather than producing stable transgenic germplasms through conventional methods such as a floral dip) also facilitates the selection of transformed cells as these have a regenerative advantage over wild-type tissues. Conversely, RNA interference could be exploited to temporarily silence suppressors of organogenesis (either by introducing chemical-inducible constructs that express antisense or hairpin templates or by transient delivery of double-stranded RNA through agroinfiltration or infection with recombinant viral vectors) because permanent gene knockouts can be detrimental to normal development and they are hard to establish in polyploid crops [173,174,175]. For example, cytokinin hypersensitivity caused by the RNAi knockdown of *CARBOXYL-TERMINAL DOMAIN PHOSPHATASE-LIKE 4* (*CPL4*) was already shown to boost de novo shoot organogenesis from root explants in *Arabidopsis* [176]. Next, further investigation is required to probe the role of epialleles in shaping phenotypic variation. Recent advances in sequencing technologies, including RNA-seq, ChIP-seq, and MethylC-seq, along with the availability of more than 1000 genomes, methylomes and transcriptomes for natural *Arabidopsis* accessions, will help to shed light on this matter by enabling accurate association of genetic, epigenetic, and transcriptional effects underlying the differential organogenic potential between and within species. Ultimately, this will contribute to the development of selectable markers to screen for highly regenerable cultivars, while a better understanding of how rate-limiting factors interact with the environment could empower targeted design of protocols for biotechnology applications in crops. Machine learning algorithms provide a complementary prospect for fine-tuning protocol development, as they allow to predict optimal requirements in terms of incubation conditions, plant growth regulators, explant source, and genotype, without the need for large-scale, time-consuming, and costly experimental trials. The potential of this approach is illustrated by recent reports on the application of artificial neural networks to model and augment somatic embryogenesis and shoot proliferation in *Chrysanthemum* and shoot regeneration in wheat by altering medium composition [177,178,179]. Similar strategies have been used to increase the production of therapeutic metabolites from in vitro cultures of *Swertia paniculata* and *Bryophyllum* [180,181]. Advanced treatments based on antibiotics, electric currents, and nanoparticles could further help to overcome recalcitrance [6]. Finally, further study is required to explain how regeneration might be linked to climate adaptation.

## Figures and Tables

**Figure 1 plants-09-01261-f001:**
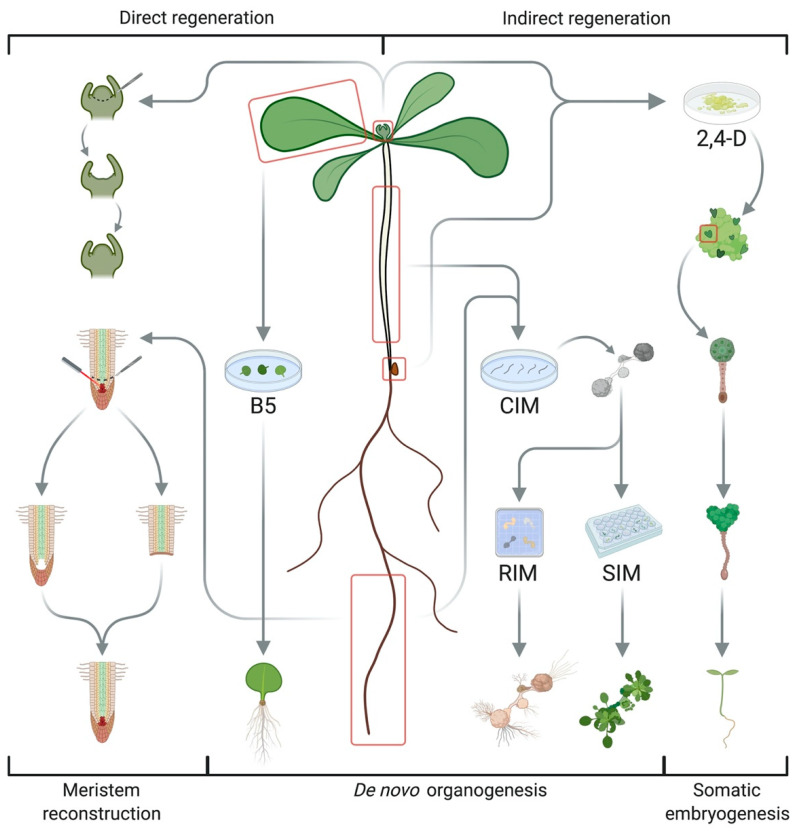
Diagram of key regeneration systems in *Arabidopsis thaliana*, including the reconstruction of shoot or root apical meristems after excision or laser ablation, adventitious rooting from excised leaves or cotyledons, de novo shoot or root organogenesis from root or hypocotyl explants, and somatic embryogenesis from immature zygotic embryos or shoot apices. This illustration was created with BioRender.com.

**Figure 2 plants-09-01261-f002:**
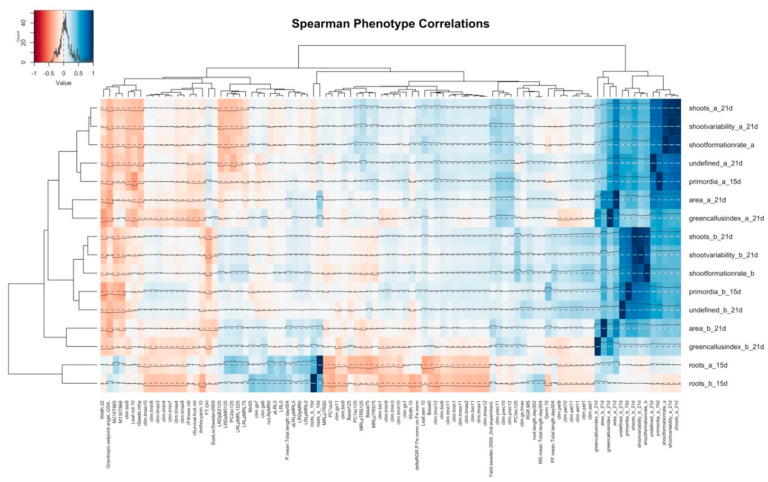
Spearman’s *rho* for pairwise comparisons between 16 regeneration traits in 170 *Arabidopsis* accessions [80] and 78 public phenotypes from AraPheno scored in at least 50 of these accessions and was significantly associated to at least one regeneration trait (FDR < 0.3) [153]. Dendrograms are based on Euclidean distances, and the legend in the upper left corner shows the color gradient and histogram of correlation coefficients. The value of the latter is also reflected by grey trace lines in the main plot. Significance is based on FDR-adjusted *p*-values from Spearman’s test (. = *p* ≤ 0.1, * = *p* ≤ 0.05, ** = *p* ≤ 0.01, *** = *p* ≤ 0.001). Accurate descriptions of all phenotypes can be found at https://arapheno.1001genomes.org/phenotypes.

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
