# Peer review of "Natural Variation in Plant Pluripotency and Regeneration"

_plants, 2020, doi:10.3390/plants9101261_

Round 1

Reviewer 1 Report

Dear Editor,

Thank you for concerning me as reviewer of the manuscript submitted to your journal, entitled: "Natural variation in plant pluripotency and regeneration" The idea of gathering a lot of information about the regeneration process and pluripotency on the basis of the wide and current world literature in this review is really good and desirable.  I think that this very good review of relevant literature would interest many researchers.

I do not think that any serious modification of this article is necessary and I recommend publishing it in "Plants" after correcting a few small suggestions, remarks mainly concerning the editing of the text:

  1. Please consider whether it is better to remove "natural" from the topic of the manuscript
  2. Keywords are not in alphabetical order
  3. Line 26-28: the definition of regeneration is too poor, is not sufficient and should be completed
  4. Line 153: I think it is good to precise such a general information to the concrete species (the cited literature 33, 34 refers to Arabidopsis)
  5. Literature and citations:
  • Line 298, 365, 576: (Lordon et al., 2020) there is not in the References
  • Line 310, 339: Motte et al. should be [77] I suppose
  • Line 395: Nishimura et al. should be [94] I suppose
  • Line should be [23] I suppose

I hope my comments will be helpful.

With best regards,

Monika Tuleja

Reviewer 2 Report

Useful review on factors controlling plants regeneration capacity. More references should be included to better underline the different regeneration processes and capacities of different more recalcitrant species/cultivars, and the study could have much more impact if better adressed to provide some possible solutions to solve the bottle neck of regeneration capacities of some of the most recalcitrant crops. 

For example, to the end of the sentence in line 44 some references on such differences should be added, see few example of recalcitrant plants:  

Peach  Sabbadini S., Ricci A., Limera C., Baldoni D., Capriotti L., e Mezzetti B., 2019. Factors Affecting the Regeneration, via Organogenesis, and the Selection of Transgenic Calli in the Peach Rootstock Hansen 536 (Prunus persica × Prunus amygdalus) to Express an RNAi Construct against PPV Virus. Plants 20198(6), 178; https://doi.org/10.3390/plants8060178 and other citation by Ricci et al reported below.

Strawberry – blueberry : Cappelletti R., Sabbadini S., Mezzetti B., 2016. The use of TDZ for the efficient in vitro regeneration and organogenesis of strawberry and blueberry cultivars. Scientia Horticulturae, 207:117–124 doi:10.1016/j.scienta.2016.05.016

Landi L., Mezzetti B., 2006. TDZ, auxin and genotype effects on leaf organogenesis in Fragaria. Plant Cell Rep., 25(4):281-8.

Grape -  Mezzetti B., Tiziana Pandolfini, Oriano Navacchi, Lucia Landi, 2002. Genetic transformation of Vitis vinifera via organogenesis. BMC Biotechnology 2002, 2:18 : http://www.biomedcentral.com/1472-6750/2/18.

Other revisions:

Line 72 – I suggest to name the adventitious rooting as rhizogenesis

Line 82 I suggest to name the shoot organogenesis as caulogenesis

Lines 122-123 and 332 – effects on ETH inhibitor on plant regeneration should be also reported – see example paper peach ETH inhibitor (Angela Ricci, Luca Capriotti, Bruno Mezzetti, Oriano Navacchi and Silvia Sabbadini, 2020. Adventitious Shoot Regeneration from In Vitro Leaf Explants of the Peach Rootstock Hansen 536. Plants 2020, 9(6), 755; https://doi.org/10.3390/plants9060755).

Line 138 – not clear if 3.3 paragraph, and the other following, are referred to all regeneration process or only to organogenesis - rhyzogenesis, indicating CIM as starting cells, and not embryogenesis.  

Lines 183, 202 and in general – do you think that a RNAi strategy can be adopted to silence genes supressing the regeneration process in recalcitrant crops? Comments about this could be helpful. In lines 370-376 you reported only genes probably promoting regeneration, but a similar study could be done in order to identify genes supressing regeneration. It can be better developed what reported in lines 519 – 520

Lines 298, 365, 432, 508, 576 - check reference: (Lardon et al., 2020).

Line 451 4.2.2. Dicot crops – only tomato – it is well know the large difference in regeneration capacities of annual and perennial dicot crops – generally perennial are more recalcitrant (fruit crops are not mentioned, many papers are on grape, strawberry and other - see Ricci, A.; Sabbadini, S.; Prieto, H.; Padilla, I.M.; Dardick, C.; Li, Z.; Scorza, R.; Limera, C.; Mezzetti, B.; Perez-Jimenez, M.; Burgos, L.; Petri, C. Genetic Transformation in Peach (Prunus persica L.): Challenges and Ways Forward. Plants 2020, 9, 971. https://doi.org/10.3390/plants9080971 summarizing 30 years of work in attempting regeneration in peach), even if also some annual are quite recalcitrant (eg. Beans).

Line 546 5.2. Source of the explants – explants competence to regeneration can be induced – see example of the meristematic bulks induced in grape and used also for other crops (Mezzetti B., Tiziana Pandolfini, Oriano Navacchi, Lucia Landi, 2002. Genetic transformation of Vitis vinifera via organogenesis. BMC Biotechnology 2002, 2:18 : http://www.biomedcentral.com/1472-6750/2/18).

Line 653 6. Implications and future perspectives – I don’t believe that selectable markers associated to regeneration can be so helpful to solve problems for crops/cvs recalcitrant regeneration. But more important to decipher the molecular mechanisms inducing/supressing regeneration process so to solve the regeneration capacities of recalcitrant crops. This remains the major limiting factor for the application of NBTs to many worldwide important crops.  

I suggest adding a figure differentiating the main regeneration processes: embryogenesis, caulogenesis, rihyzogenesis and callogenesis. Many are already available but it could be really useful to have an additional one linked to another figure/table reporting the main genetic/molecular/genes factor inducing/suppressing regeneration.

Reviewer 3 Report

This manuscript can be categorized as one of the best review papers in the field of in vitro culture.
The authors have nicely reviewed different molecular aspects of regeneration systems.
However, there are some similar papers in this field and some of them publish in Plants journal. I strongly recommend using these papers in your manuscript.
Plants 2019, 8, 38; doi:10.3390/plants8020038
Plants 2020, 9, 702; doi:10.3390/plants9060702
Regeneration. 2017;4:201–216; doi:10.1002/reg2.91
Also, recently, some papers published about using machine learning algorithms for comprehending regeneration systems. I suggest adding this strategy (machine learning algorithm) as a promising approach for understanding and modeling regeneration systems. You can use the following papers:
Plant Methods 2020, 16: 112; doi:10.1186/s13007-020-00655-9
Antioxidants 2020, 9, 210; doi:10.3390/antiox9030210
Appl. Sci. 2020, 10, 5370; doi:10.3390/app10155370
BMC Plant Biology (2020) 20:225; doi:10.1186/s12870-020-02410-7
Sci Rep 2019; 9, doi:10.1038/s41598-019-54257-0
Also, I suggest preparing a table for showing previous molecular studies and summarize previous studies in the table.

Round 2

Reviewer 2 Report

Authors have not taken in consideration some of the comments proposed so to better adress the paper. The concept of useing molecolar breeding for selecting some more easy to regenerate plants can be applied only to some crops and still have a high limitation. For publication this should be better clarified in the manuscript.